# Combined Modality Therapy Based on Hybrid Gold Nanostars Coated with Temperature Sensitive Liposomes to Overcome Paclitaxel-Resistance in Hepatic Carcinoma

**DOI:** 10.3390/pharmaceutics11120683

**Published:** 2019-12-15

**Authors:** Hongyan Zhu, Weili Han, Ye Gan, Qiaofeng Li, Xiaolan Li, Lanlan Shao, Dan Zhu, Hongwei Guo

**Affiliations:** 1College of Pharmacy, Nantong University, Nantong 226001, China; amy600@ntu.edu.cn (H.Z.); pisces_lier@163.com (W.H.); 1823310018@stmail.ntu.edu.cn (Y.G.); 1923310020@stmail.ntu.edu.cn (L.S.); 2Key Laboratory of Longevity and Aging-related Diseases of Chinese Ministry of Education & Center for Translational Medicine, Guangxi Medical University, Nanning 530021, China; liqiaofeng@stu.gxmu.edu.cn (Q.L.); lixiaolan@stu.gxmu.edu.cn (X.L.); 3School of Preclinical Medicine, Guangxi Medical University, Nanning 530021, China; 4College of Pharmacy, Guangxi Medical University, Nanning 530021, China

**Keywords:** gold nanostars, temperature sensitive liposome, siCOX-2, paclitaxel-resistance, hepatic carcinoma

## Abstract

In this study, we prepared gold nanostar (GNS) composite nanoparticles containing siRNA of cyclooxygenase-2(siCOX-2) that were modified by tumor targeting ligand 2-deoxyglucose (DG) and transmembrane peptide 9-poly-*D*-arginine (9R) to form siCOX-2(9R/DG-GNS). Paclitaxel loaded temperature sensitive liposomes (PTX-TSL) were surface-modified to produce PTX-TSL-siCOX-2(9R/DG-GNS) displaying homogeneous star-shaped structures of suitable size (293.93 nm ± 3.21) and zeta potentials (2.47 mV ± 0.22). PTX-TSL-siCOX-2(9R/DG-GNS) had a high thermal conversion efficiency under 808 nm laser radiation and a superior transfection efficiency, which may be related to the targeting effects of DG and increased heat induced membrane permeability. COX-2 expression in HepG2/PTX cells was significantly suppressed by PTX-TSL-siCOX-2(9R/DG-GNS) in high temperatures. The co-delivery system inhibited drug-resistant cell growth rates by ≥77% and increased the cell apoptosis rate about 47% at elevated temperatures. PTX-TSL and siCOX-2 loaded gold nanostar particles, therefore, show promise for overcoming tumor resistance.

## 1. Introduction

Paclitaxel is widely administered to patients with breast and ovarian cancer and is beneficial to the treatment of lung and neck cancer [1,2]. Although paclitaxel is an effective chemotherapeutic agent, several mechanisms of paclitaxel resistance that impede its clinical efficacy have been reported [3,4,5,6], including the overexpression of P-glycoprotein (P-gp), resulting in increased drug efflux from tumour cells [7,8,9], microtubule mutations, and the overexpression of anti- and pro-apoptotic members of the B-cell lymphoma-2 (Bcl-2) family [10,11,12]. Due to drug resistance, epothilones (paclitaxel analogues) have replaced paclitaxel for cancer therapy, but their clinical efficacy is limited. In addition, new cytotoxic drugs have been developed for paclitaxel resistant ovarian cancer, but remission rates remain as high as 15%–25% [13]. The use of agents that inhibit P-gp mediated drug efflux are another attractive strategy, but they are still under clinical trial, with their therapeutic benefits remaining unknown [14]. Therefore, new therapeutic strategies to improve paclitaxel resistance are urgently required.

Temperature-sensitive liposome (TSL) can improve antitumor drug efficacy and reduce drug side effects [15,16,17,18,19,20]. The treatment process is initiated by heating, which can be advantageous. At high temperatures, TSL increases the release of encapsulated anticancer drugs at the tumor site [21,22]. Additionally, hyperthermia (HT) (41–43 °C) can inhibit tumor cell processes, enhance blood flow [23], and increase the permeability of tumor vessels to drug delivery systems [24]. The combination of HT and TSL can improve drug delivery and efficacy [25].

Gold nanoparticles can be combined with many drugs, permitting diverse biomedical applications, including delivery systems, medical imaging, and diagnostics [26]. Gold nanostars (GNS) are a new type of gold nanoparticle with excellent physical and chemical properties. They are capable of converting light into heat when light irradiation’s wavelength coincides with the unable surface plasmon resonance (SPR) of GNS [27,28]. In addition, GNS modifications exhibit plasmons in the near-infrared (NIR) region, making GNS a photothermal treatment option that can specifically target tumor cells when their surface is further modified with target ligands. Moreover, the cutting-edge structure of GNS displays increased heat production capacity and deep penetration into the tumor tissue, minimizing tumor drug resistance [29,30,31,32,33,34,35].

Cyclooxygenase-2 (COX-2) is a key metabolite of arachidonic acid and plays a major role in tumorigenesis, apoptosis, and angiogenesis [36,37,38]. Numerous studies have reported that high COX-2 expression is an early marker of tumor development [39,40,41,42,43,44,45]. In addition, COX-2 is involved in inducing multi-drug resistance in tumors through increased Bcl2 expression [46] and upregulating multi-drug resistance related proteins [47,48]. Hence, COX-2 silencing using small interfering RNAs (siRNAs) can suppresses drug resistance in tumor cells.

Here, we prepared siCOX-2-loaded composite GNS modified with 2-deoxyglucose (DG) and 9 polyarginine (9R) to form siCOX-2(9R/DG-GNS). A co-delivery system of paclitaxel-loaded temperature sensitive liposomes (PTX-TSLs) and siCOX-2(9R/DG-GNS) was further constructed to form PTX-TSL-siCOX-2(9R/DG-GNS) (Figure 1). The photothermal effects of GNS permitted phase transition temperatures of the liposomes. After heating, the PTX was simultaneously released and siCOX-2(9R/DG-GNS) was exposed. siCOX-2(9R/DG-GNS) was then taken up by drug-resistant tumor cells through the active targeting role of DG. The co-delivery of PTX-TSL-siCOX-2(9R/DG-GNS) was thus predicted to overcome paclitaxel resistance in cancer cells.

## 2. Materials and Methods

### 2.1. Materials

Polyethylene glycol (PEG, relative molecular mass: 2000, 98.6%), 1, 2-distearoyl-sn-glycero-3-phosphoethanolamine-PEG-SH (DSPE-PEG-SH, relative molecular mass: 5000, 97.0%) and 9 polyarginine (9R, 98.0%) were synthesized by GL Biochem Ltd. (Shanghai, China). 2-amino-2- deoxy-D-glucose hydrochloride (DG· HCl, 97.0%), n-hydroxy-succinamide (NHS, 99.0%), dicyclohexyl carbodiimide (DCC, 99.0%), 1-(3-Dimethylaminopropyl)-3-ethylcarbodiimide hydrochloride (EDC, 99.0%) and *L*-cysteine (Cys) were purchased from Aladdin Reagents (Shanghai, China). 3-(4, 5-dimethylthiazol-2-yl)-2, 5-diphenyltetrazolium bromide (MTT), RPMI 1640, DMEM high-glucose medium, fetal bovine serum (FBS), penicillin, streptomycin and trypsin were purchased from Gibco (New York, NY, USA). Paclitaxel (PTX, 99.0%) was purchased from Jiangsu yew pharmaceuticals (Wuxi, China). 1, 2-dipalmitoyl-sn-glycero-3-phosphocholine (DPPC, 99.0%), 1-stearoyl-2-hydroxy-sn-glycero-3-phosphocholine (MSPC, 99.0%) and cholesterol were purchased from Avanti (Alabaster, AL, USA). COX-2 siRNA (sense strand, 5′-AAC UGC UCA ACA CCG GAA Utt-3′, antisense strand 5′-AUU CCG GUG UUG AGC AGU Utt-3′) and the non-specific control siRNA (NC siRNA) (sense strand, 5′-UUC UCC GAA CGU GUC ACG U-3′, antisense strand 5′-ACG UGA CAC GUU CGG AGA A-3′) were purchased from Biomics Biotechnologies (Shanghai, China). The primary antibodies including anti-COX-2 from Cell Signaling Technology (Danvers, MA, USA) and anti-GAPDH from KANCHEN (Shanghai, China). The secondary antibodies included IRDye 680-labeled secondary antibodies from Santa Cruz Biotech (Santa Cruz, CA, USA). Other experimental reagents were purchased from the Shanghai Chemical Reagent Company (Shanghai, China).

### 2.2. Synthesis of the DG-PEG-Cys-9R Complex

#### 2.2.1. DG-PEG Synthesis

The first step was to protect the amino group of NH_2_-PEG-COOH by di-tert-butyl pyrocarbonate ((Boc)_2_O). NH_2_-PEG-COOH was dissolved in Dichloromethane and the catalyst 4-dimethylaminopyridine (DMAP) was added. After stirring for 5 min, (Boc)_2_O was added dropwise under ice bath conditions and stirred for 30 min. The solution was purified by dialysis (MW 2000) after stirring overnight at room temperature. The ninhydrin reaction was used to verify the successful protection by (Boc)_2_O.

Step two was the carboxyl group of NH_2_-PEG-COOH with Boc-protected amine reacting with DG. Firstly, the carboxyl group of NH_2_-PEG-COOH was catalytically activated by a DCC/NHS system (molar ratio of PEG:DCC:NHS = 1:1.8:2.25) in anhydrous dimethylformamide (DMF) solution with continuous stirring overnight at room temperature. Centrifugation was performed to remove byproducts before DG∙HCl was dissolved in the activated PEG solution with stable stirring for 4 h (molar ratio of PEG:DG = 1:0.8). The obtained DG-PEG complex was then dissolved in a mixed solution of dichloromethane and trifluoroacetic acid with constant agitation (volume ratio: 7:3) to remove the Boc-protection on amino group of NH_2_-PEG-COOH. The DG-PEG complex was then purified by dialysis (MW 2000) and obtained by lyophilization.

#### 2.2.2. Synthesis of DG-PEG-Cys Complex

The amine group of L-Cysteine (Cys) was protected by (Boc)_2_O, in a similar fashion to the amino group of NH_2_-PEG-COOH by (Boc)_2_O. The carboxyl group of Cys was activated by a DCC/NHS catalyst system (molar ratio of PEG:DCC:NHS = 1:1.8:2.25). The prepared DG-PEG complex was reacted with the activated solution with stable stirring for 4 h. The products were dissolved in a mixed solution of dichloromethane and trifluoroacetic acid (volume ratio of 7:3). Finally, DG-PEG-Cys was purified by dialysis (MW 2000) and obtained via rotary evaporation. NH_2_-PEG-OCH_3_ without DG modifications was used for the reaction with the Cys to obtain the Cys-PEG-OCH_3_ as a control.

#### 2.2.3. Synthesis of the DG-PEG-Cys-9R Complex

EDC and NHS (molar ratio of 9R:EDC:NHS = 1:1.2:1.5) was used to activate the carboxy group of 9R (the terminal amine group protected by 9-fluorenylmethyloxycarbony (Fmoc)) for 4 h at room temperature. Then, the activated carboxy group of 9R was allowed to bond with the amine group of DG-PEG-Cys with constant stirring for 4 h at room temperature. The synthetic products were dissolved in a mixed solution of dimethyformamide (DMF) solution and 20% piperidine (volume ratio of 7:3) to divest Fmoc. Finally, the DG-PEG-Cys-9R complex was obtained (molar ratio of 9R:DG-PEG-Cys = 1:1.5). The experimental control group H_3_CO-PEG-Cys-9R was the same as above.

### 2.3. Preparation of siCOX-2(9R/DG-GNS)

#### 2.3.1. Preparation of GNS

GNS was prepared via the seed growth method. Chloroauric acid (HAuCl_4_) solution (10 mmol/L), 9.75 mL H_2_O, 1 mol/L hydrochloric acid and 100 μL seed solution were successively added to a 10 mL reaction bottle in a 25 °C water bath. Silver nitrate (AgNO_3_) (2 mol/L, 100 μL) and ascorbic acid (AA) (100 mmol, 50 μL) were also added. GNS solution was obtained after stirring for 30 s.

#### 2.3.2. Preparation of 9R/DG-GNS

DG-PEG-Cys-9R (0.05 mmol) was fully dissolved in H_2_O and 0.04 mmol GNS (10 mL) was added with stirring for 12 h at room temperature. 9R/DG-GNS was obtained and modified by 9R and DG simultaneously. Similarly, GNS reacted with H_3_CO-PEG-Cys-9R to obtain 9R-GNS as a control.

#### 2.3.3. Preparation of siCOX-2(9R/DG-GNS)

9R/DG-GNS and siCOX-2 (molar ratio of 1:1) were mixed and incubated on ice for 45 min to obtain siCOX-2-loaded 9R/DG-GNS(siCOX-2(9R/DG-GNS)). siCOX-2(9R-GNS) was obtained as control using the same method.

### 2.4. Preparation of PTX-TSL-siCOX-2(9R/DG-GNS)

DPPC (90 mg), MSPC (5 mg), DSPE-PEG-SH (10 mg), cholesterol (10 mg) at molar ratios of 9:0.5:1:1 and PTX (9 mg) (molar ratio of paclitaxel:phospholipid = 1:10) were dissolved in 5 mL of chloroform which was volatilized by vacuum evaporation forming a thin film on the bottle wall. The remaining chloroform was removed by rotary evaporation. Phosphate buffer saline (PBS) (21 mL) was added to hydrate for 2 h at 60 °C and siCOX-2(9R/DG-GNS) was added to the aqueous solution for further hydration overnight. Mixtures were sonicated for 10 min and filtered (0.22 μm membrane). The co-delivery PTX-loaded and siCOX-2(9R/DG-GNS) (PTX-TSL-siCOX-2(9R/DG-GNS)) system were obtained. PTX-TSL without siCOX-2(9R/DG-GNS) was obtained as control.

### 2.5. Characterization of PTX-TSL-siCOX-2(9R/DG-GNS)

#### 2.5.1. Characterization of DG-PEG-Cys-9R Complex

The synthesis of DG, HOOC-PEG-NH_2_, DG-PEG-Cys, H_3_CO-PEG-NH_2_ and H_3_CO-PEG-Cys was identified by ^1^H nuclear magnetic resonance (^1^H NMR) spectra. Using deuterium dimethyl sulfoxide (DMSO) as the solvent DG, HOOC-PEG-NH_2_, DG-PEG-Cys, H_3_CO-PEG-NH_2_ and H_3_CO-PEG-Cys with a mass concentration of 2 mg/mL was prepared for the assessment of its nuclear magnetic spectrum.

SDS-PAGE was performed on acrylamide gels (7.5% resolving gel, 4% stacking gel) to identify the DG-PEG-Cys-9R and H_3_CO-PEG-Cys-9R systems.

#### 2.5.2. Physicochemical Property of Hybrid Gold Nanostars

The Fourier transform infrared (FTIR) spectra of GNS, 9R/DG-GNS, TSL and TSL-(9R/DG-GNS) were recorded using a FTIR spectrometer (Thermo Fisher Scientific, Waltham, MA, USA). We also recorded the FTIR spectra of DG, HOOC-PEG-NH_2_, DG-PEG and DG-PEG-Cys as control groups.

The hydrodynamic diameter, PDI, and zeta potential of hybrid GNS were measured using a Malvern Zetasizer Nano ZS90 (Malvern Instrument Ltd., Worcestershire, UK). The morphologies of the nanoparticles were imaged via JEM-1011transmission electron microscope (TEM) (JEOL, Tokyo, Japan). UV-VIS-NIR spectra were obtained on a 754 PC spectrometer (Jinghua Instruments, Shanghai, China).

The photothermal potential of prepared PTX-TSL-siCOX-2(9R/DG-GNS) was investigated by NIR laser irradiation (808 nm) (LEO-Photoelectric, Shenzhen, China). The temperature was monitored with a thermometer every 30 s for 10 min. The photothermal performances of PTX-TSL-siCOX-2(9R/DG-GNS) under various laser intensities and different concentrations of samples were investigated respectively. In addition, GNS and GNS composite carriers were irradiated for 10 min and photothermal performances were monitored. We also investigated the photothermal performances of 9R, DG, HOOC-PEG-NH_2_, 9R/DG and 9R/DG-GNS alone, which served as negative controls. To further demonstrate the photothermal stability of PTX-TSL-siCOX-2(9R/DG-GNS), the photothermal efficiency was tested for five cycles of NIR laser irradiation and cooling down.

To assess the binding ability of hybrid gold nanostars with siCOX-2, different siCOX-2 ratios were assessed (molar ratio of 1:0, 1:1, 1:5, 1:10, 1:20, 1:40, 1:60) at 4 °C for 45 min. Prepared samples were run on 4% agarose gels at 120 V for 20 min and imaged.

To examine the drug-loading properties of temperature sensitive liposomes in detail, high-performance liquid chromatography (HPLC) (Shimadzu, Kyoto, Japan) was used to detect the PTX content of PTX-TSL-siCOX-2(9R/DG-GNS). The chromatographic conditions were as follows: detection wavelength: 227 nm; column: C18; mobile phase: water-acetonitrile-methanol (38:40:22); injection volume: 20 μL; flow rate: 1 mL/min, column temperature: 25 °C. A PTX standard curve was constructed by dissolving the temperature-sensitive liposome solution containing PTX and filtering through 0.45 μm membranes. Drug loading and encapsulation efficiency were calculated under the above liquid phase conditions and estimated from the following formula:$$\mathrm{Drug}\;\mathrm{loading}\;\mathrm{efficiency}\%\;=\;\frac{\mathrm{weight}\;\mathrm{of}\;\mathrm{drug}\;\mathrm{in}\;\mathrm{liposomes}}{\mathrm{weight}\;\mathrm{of}\;\mathrm{liposomes}}\times 100\%$$
$$\mathrm{Encapsulation}\;\mathrm{efficiency}\%\;=\;\frac{\mathrm{weight}\;\mathrm{of}\;\mathrm{drug}\;\mathrm{in}\;\mathrm{liposomes}}{\mathrm{weight}\;\mathrm{of}\;\mathrm{drug}\;\mathrm{added}}\times 100\%.$$

The phase change temperature of the temperature sensitive liposome were measured by a differential scanning calorimeter (DSC) (Netzsch, Selb, Bavaria, Germany).

The dialysis method was used to detect the release properties of temperature-sensitive liposomes under different temperature conditions. PTX-TSL and PTX-TSL-siCOX-2(9R/DG-GNS) were added to dialysis bags and 250 mL of 30% ethanol in PBS was used as the dialysis medium. The dialysis media (1 mL) was collected at 0, 1, 5, 10, 20, 40, 60 min and added to an equal volume of PBS. The amount of PTX released in the dialysis medium was determined by HPLC. The release percentage of PTX was calculated as the formula of weight (cumulative release of PTX)/weight (total PTX).

To investigate the stability of the co-delivery system, the particle sizes and PDI of PTX-TSL-siCOX-2(9R/DG-GNS) were detected at different weeks: 1, 2, and 3 at 4 °C.

### 2.6. Cell Experiments

#### 2.6.1. Cell Biocompatibility

MTT assay was performed to assess the cytotoxicity of free GNS and GNS composite carriers. Briefly, Human Umbilical Vein Endothelial Cells (HUVECs) were cultured in RPMI 1640 medium containing 10% (v/v) FBS, streptomycin (100 μg/mL) and penicillin (100 U/mL) at 37 °C in 5% CO_2_. Cells were seeded at 5000 cells/well in 96-well plates (Corning Inc., New York, NY, USA) and allowed to grow into 70%–80% confluency. Then, cells were treated with GNS and GNS composite carriers in RPMI 1640 medium (200 μL) at concentrations ranging from 0.01 to 1000 μmol/L for 48 h. Cells were incubated in MTT reagent for 4 h and 150 μL DMSO was added to dissolve the intracellular formazan crystals. Optical densitie (OD) at 490 nm was detected using an Infinite F200 (TecanInc, Zurich, Switzerland). Cell viability (%) was calculated as the OD_treated_/OD_control_ × 100%, where OD_treated_ represents the OD in the presence of GNS and GNS composite carrier while OD_control_ is the absorbance of cells not treated with GNS and the GNS composite carrier.

#### 2.6.2. In Vitro Cellular Uptake Assays

PTX-resistant HepG2/PTX cells were seeded in a 35 mm^2^ petri dishes (MatTek, Ashland, MA, USA) at a density of 1.0 × 10^5^ cells/well and grown for 24 h at 37 °C in 5% CO_2_. The nucleus was stained by 4, 6-Diamidino-2-Phenylindole (DAPI) overnight according to the manufacturer’s protocol. Cells were transfected with Carboxyfluorescein (FAM) labeled siRNA delivery systems (siCOX-2(GNS), siCOX-2(9R-GNS), siCOX-2(9R/DG-GNS), PTX-TSL-siCOX-2(9R/DG-GNS)). The final concentration of FAM-siRNA was 50 nM. After incubation for 4 h, the medium was discarded and the cells were rinsed 3 times with PBS (PH = 7.4). The uptake of siRNA and GNS complexes was assessed through imaging on a Leica TCS SP5 confocal laser-scanning microscope (CLSM) (Leica, Buffalo Grove, IL, USA). FAM-labeled siRNA had an excitation wavelength of 492 nm. All images were quantified using ImageJ software (National Institutes of Health; Bethesda, MD, USA).

#### 2.6.3. Analysis of In Vitro Gene Silencing

COX-2 expression in PTX- resistant HepG2 cells was assessed by a Western blot analysis following treatment with different concentrations of free siCOX-2, GNS complexes (siCOX-2/9R-GNS, siCOX-2(9R/DG-GNS) and siCOX-2(9R/DG-GNS) (43 °C) (15 min)) and GNS-PTX complexes (PTX-TSL-siCOX-2(9R/DG-GNS), PTX-TSL-siNC(9R/DG-GNS) (43 °C) (15 min) and PTX-TSL-siCOX-2(9R/DG-GNS) (43 °C)). Briefly, cells were lysed in lysis buffer containing (50 mmol/L Tris-HCl, 100 mmol/L NaCl, 1 mmol/L EDTA, 3 mmol/L Na_3_VO_4_, 20 mmol/L NaF, 1 mmol/L PMSF, supplemented with 1% Nonidet P-40 and protease inhibitor cocktail). BCA protein quantification assay was performed. Then, the samples with known concentrations were all dispersed in a loading buffer and heated at 95 °C for 5 min. The proteins of different samples (30 μg) were separated on SDS-PAGE (10%) gels for electrophoresis and transferred to nitrocellulose membranes. The membranes were blocked in 5% non-fat milk powder for 2 h with slight shaking at room temperature. Then, the membranes were washed with TBST buffer (996 mL H_2_O, 6.057g Tris, 8.5 g NaCl, 0.05% Tween 20, pH 7.5–7.6). Membranes were incubated overnight at 4 °C with anti-COX-2 (1:1000) and anti-GAPDH (1:1000) antibodies and washed three times in TBST for 30 min. The GAPDH membrane and COX-2 membrane were then incubated with IRDye 680-labeled mouse and rabbit secondary antibodies (1:15,000) at room temperature for 2 h in the dark, respectively. The expression of DAPDH and COX-2 were assessed on an Odyssey CLx Western blot detection system (LI-COR Biosciences, Lincoln, NE, USA).

#### 2.6.4. Cell Growth and Anti-Tumor Drug Resistance

MTT assays were used to evaluate the cytotoxicity of different composite carriers [49,50]. Briefly, PTX-resistant HepG2 cells (5000 cells per well) were plated into 96-well flat-bottomed microtiter plates to 70%–80% confluence prior to use. Then, the cells were treated with DMEM containing siNC(9R/DG-GNS), siCOX-2(GNS), siCOX-2(9R-GNS), siCOX-2(9R/DG-GNS), PTX-TSL-siNC(9R/DG-GNS) and PTX-TSL-siCOX-2(9R/DG-GNS) (final PTX concentration was 7 nM) for 48 h. The final concentrations of siRNA (siNC and siCOX-2) were 25 nM, 50 nM and 100 nM. After 48 h incubation, the medium was removed and cells were incubated in MTT reagent (20 μL, 5 mg/mL) for 4 h. Then, 150 μL DMSO was added to dissolve the MTT formazan crystal for 10 min and the ODs were detected at 490 nm according to manufacturer’s protocol. Cell viability (%) was defined as the OD_treated_/OD_control_ × 100%, where OD_treated_ was the OD obtained in the presence of GNS and GNS composite carriers and the OD_control_ was the absorbance of untreated cells.

#### 2.6.5. Cell Apoptosis Assay

PTX-resistant HepG2 cells were seeded in a 24-well plate at a density of 5 × 10^4^ cells per well. After 24 h of incubation, the medium was replaced with the medium containing siNC(9R/DG-GNS), siCOX-2(GNS), siCOX-2(9R-GNS), siCOX-2(9R/DG-GNS), PTX-TSL-siNC(9R/DG-GNS), PTX-TSL-siCOX-2(9R/DG-GNS) respectively (the concentration of siRNA was 50 nM; the concentration of PTX was 7 nM. The heating groups were incubated at 43 °C in 5% CO_2_ for 15 min). After being treated for 4 h, the cells were collected by centrifugation and washed once with PBS. After cell suspension in 200 μL binding buffer, staining was performed with propidium iodide (PI) and annexin V-FITC for 20 min at room temperature. The fluorescence intensities of cells were recorded by flow cytometry.

### 2.7. Statistical Analysis

A statistical analysis was performed using GraphPad Prism software (Version 5.01) (GraphPad Software, San Diego, CA, USA). A student’s *t*-test was used to analyze the differences between two independent groups. A *P* value at 0.05 or less was considered statistically significant.

## 3. Results and Discussion

### 3.1. Characterization of DG-PEG-Cys-9R, 9R/DG-GNS and TSL-9R/DG-GNS

The chemical structure of PTX-TSL-siCOX-2(9R/DG-GNS) and the preparation process are illustrated in Scheme 1. The ^1^H NMR spectra of DG-PEG-Cys demonstrated that 3.67–3.69 ppm and 4.02 ppm belonged to DG; 3.51 ppm and 3.34 ppm belonged to CH_2_ of PE; 1.49–1.74 ppm and 5.57–5.59 ppm belonged to Cys and 2.50 ppm corresponded to DMSO peak. The ^1^H NMR spectra from H3CO-PEG-Cys demonstrated that 3.34 ppm and 3.50 ppm belonged to CH_2_ of PEG compared; 1.23–1.70 ppm and 5.56–5.58 ppm belonged to Cys and 2.50 ppm corresponded to DMSO peak. These data confirm the successful linkage of DG-PEG-Cys and H3CO-PEG-Cys.

The successful conjugation of DG-PEG-Cys and 9R, H_3_CO-PEG-Cys and 9R was confirmed by gel electrophoresis. Figure 1A shows that DG-PEG-Cys-9R (a) and H_3_CO-PEG-Cys-9R (c) had a higher molecular weight, confirming their successful linkage.

The FTIR spectra of DG displayed a 3291 cm^−1^ for O-H; C-O for 1034 cm^−1^. Moreover, the FTIR spectrum of HOOC-PEG-NH_2_ revealed that the emerging peak at 1111 cm^−1^ was the C-O-C in PEG, and the peak at 2885 cm^−1^ was attributed to C-H in PEG; the peak at 1734 cm^−1^ was due to the presence of C=O in PEG. After the reaction of DG with HOOC-PEG-NH_2_, the peaks at 3329 cm^−1^ and 1577 cm^−1^ could be attributed to N-H, and the peak at the 1627 cm^−1^ was assigned to C=O, indicating the formation of amide in DG-PEG. After the reaction of Cys with DG-PEG, the peak could be found at 1629 cm^−1^, which could be attributed to C=O. Moreover the 1572 cm^−1^ and 3330 cm^−1^ were assigned to N-H, which indicated the successful formation of amide in DG-PEG-Cys. Compared to the FTIR spectrum (Figure 1B) of free GNS, 9R/DG-GNS displayed a 1085 cm^−1^ stretching band for C-O-C of PEG; 1685 cm^−1^ for the C=O of the amido bond; 3328 cm^−1^ as the main overlapping peak on the stretching band for N-H of the amido bond; and 2928 cm^−1^ for the CH_2_ of PEG, indicating the formation of 9R/DG-GNS. The FTIR spectrum of TSL revealed that the emerging peak at 1092 cm^−1^ was the stretching band for C-O-C of DSPE-PEG in TSL; the peaks at 2919 cm^−1^ and 2850 cm^−1^ were attributed to the presence of methylene and methyl groups in the PEG molecules; and the peak at 1738 cm^−1^ was due to the presence of DPPC in the TSL. The FTIR spectra of TSL-(9R/DG-GNS) showed that 3419 cm^−1^ was an overlapping peak on the symmetrical stretching vibration of the O-H of DG and the N-H of the amido bond; while the peaks at 2918 cm^−1^ and 2850 cm^−1^ were attributed to the presence of methylene and methyl groups in the PEG molecules. The peak at 1739 cm^−1^ was attributed to the presence of DPPC in the PTX-TSL. These results indicate that the TSL was attached to the surface of 9R/DG-GNS, confirming the successful construction of TSL-(9R/DG-GNS).

### 3.2. Particle Size, Zeta Potentials, and Morphology of the Nanocarriers

The particle sizes of GNS, 9R-GNS, 9R/DG-GNS, siCOX-2(9R-GNS), siCOX-2(9R/DG-GNS), PTX-TSL and PTX-TSL-siCOX-2 (9R/DG-GNS) were 57.23 ± 3.42 nm, 89.41 ± 5.53 nm, 199.12 ± 3.91 nm, 203.26 ± 6.21 nm, 231.48 ± 5.27 nm, 93.56 ± 5.17 nm and 293.93 ± 3.21 nm, respectively (Figure 1C, Table 1). The functionalization of 9R, DG and PTX-TSL on the surface of GNS explained the increased size of the hybrid nanoparticles in aqueous solution. The zeta potentials of GNS, 9R-GNS, 9R/DG-GNS, siCOX-2(9R-GNS), siCOX-2(9R/DG-GNS), PTX-TSL and PTX-TSL-siCOX-2(9R/DG-GNS) were 0.12 ± 0.17 mV, 19.79 ± 0.16 mV, 10.85 ± 0.25 mV, 0.26 ± 0.27 mV, 0.16 ± 0.62 mV, −1.78 ± 0.41 mV and −2.47 ± 0.22 mV, respectively (Figure 1D, Table 1). The 9R provided a positive surface charge and could adsorb negative siCOX-2. PEG modifications shielded the positive charge of GNS and 9R, which helped reduce the toxicity of the co-delivery system. The PDI of each sample group was ≤0.3, demonstrating an improved distribution of the sample particle sizes.

GNS composite carriers were well dispersed as individual nanoparticles and star-shaped based on TEM images (Figure 2). The morphological features of 9R-GNS, 9R/DG-GNS and siCOX-2(9R/DG-GNS) did not differ from GNS. However, the thin film of PTX-TSL was clearly evident on the surface of GNS complexes after modification. The morphological features of PTX-TSL are shown in Figure 2F, in which a round and relatively uniform distribution was observed. It is worth mentioning that the TEM sizes of GNS composite carriers were measured in a dry state, which probably distorted the real sizes and sharpness. However, the hydrodynamic diameter is the real diameter under hydrated conditions that was larger than those observed by TEM. This phenomenon is commonly found in previous report [51,52].

### 3.3. Spectral Identification and Photothermal Effects of the Nanocarriers

UV-VIS-NIR absorption spectra are shown in Figure 3A. The maximum UV-VIS-NIR absorption peaks of GNS, 9R-GNS, 9R/DG-GNS, siCOX-2(9R-GNS), siCOX-2(9R/DG-GNS) and PTX-TSL-siCOX-2(9R/DG-GNS) were 785 nm, 792 nm, 796 nm, 810 nm, 815 nm and 825 nm, respectively. No absorption peak was observed near 808 nm in the non-GNS group (Appendix A). The UV-VIS-NIR absorption peaks displayed a red-shift with no changes in peak width. This confirmed that the red-shift of the UV-VIS-NIR absorption peaks was related to their enhanced particle size after the chemical modification of GNS.

As displayed in Figure 3B, PTX-TSL-siCOX-2(9R/DG-GNS) (30 μg/mL) increased temperature from 26.6 to 73.3 °C, irradiated with an 808 nm laser at 0.25–1.00 w/cm^2^. Figure 3C shows that temperature validations of PTX-TSL-siCOX-2(9R/DG-GNS) dispersion were from 23.3 to 50.6 °C under the concentration of 0–120 μg/mL with a laser at 0.5 w/cm^2^. As shown in Figure 3D and Appendix A, there was no apparent temperature increase in the negative control groups (9R, DG, HOOC-PEG-NH2, 9R-Cys-PEG-OCH_3_ and 9R-Cys-PEG-DG) after NIR laser irradiation (0.5 W/cm^2^) for 10 min. However, the temperature curves of GNS composite carriers revealed that further modification of targeted ligand caused a minimal impairment on the photothermal effect of GNS. Moreover, the photostability of PTX-TSL-siCOX-2(9R/DG-GNS) was performed under 808 nm laser every 20 min over 5 on/off cycles (Figure 3E). Apparently, the photothermal conversion effect of PTX-TSL-siCOX-2(9R/DG-GNS) demonstrated its favorable stability and promising potential as a photothermal therapy agent for tumor therapy. To avoid excessive laser power and unnecessary damage to normal tissue in subsequent animal experiments, an NIR laser at 0.5 w/cm^2^ was chosen. Then, the therapeutic temperature of 43 °C for the following anti-tumor experiments was reached with PTX-TSL-siCOX-2(9R/DG-GNS) (30 μg/mL) under 0.5 w/cm^2^.

### 3.4. Analysis of the siCOX-2 Encapsulation

The siCOX-2 binding abilities of carriers based on 9R-GNS or 9R/DG-GNS were analyzed by gel shift assay [53,54]. As shown in Figure 4A, the composite nanoparticle of GNS complexes (9R-GNS, 9R/DG-GNS) and siCOX-2 was prevented from moving towards the positive electrode in 4% agarose gels. When siCOX-2 was combined with the nanocarriers at the correct ratio, the positively charged ligand neutralized the negative charge of siCOX-2, and its movement towards the positive electrode was inhibited [55]. Then, the GNS complex was coated with PTX-TSL by mercapto coordination, which protects siCOX-2 and increases its stability. When the GNS core was coated by PTX-TSL, we found that the construction of PTX-TSL-siCOX-2(9R/DG-GNS) can be formed and optimized at the molar ratio of (molar ratio of siCOX-2/PTX-TSL-(9R/DG-GNS) was 1:40 or 1:60).

### 3.5. Detection of Drug Loading and Release Capacity of PTX-TSL

The PTX peak areas of the concentration gradients were determined by HPLC. The PTX concentration was used as an abscissa and the peak area was used as the ordinate for standard curve construction: y = 26111 x ± 26707 (*R^2^* = 0.9986). The drug loading and encapsulation efficiency of PTX-TSL-(9R/DG-GNS) was calculated as 7.2% ± 1.19 and 92.98% ± 1.07, respectively.

The phase changes of the temperature sensitive liposomes by DSC were 42.6 °C. Both 37 °C and 43 °C were selected as the drug release conditions. As shown in Figure 4B, the liposomes were stable at 37 °C and the released PTX-TSL and PTX-TSL-siCOX-2(9R/DG-GNS) were ≤10% after 60 min. This was primarily due to the temperature being below the phase transition temperature (Tm). The temperature-sensitive liposome membranes were in a densely packed colloidal state through which drug diffusion was low. Surprisingly, PTX-TSL and PTX-TSL-siCOX-2(9R/DG-GNS) had rapid release kinetics in the first 20 min which remained at ~80% under heating conditions (43 °C) (Figure 4B). This was because at 42 °C, the outer membranes underwent phase transition, became permeable, and the encapsulated drug was rapidly released. Temperatures of 43 °C were slightly higher than the phase transition temperatures (42.6 °C) of temperature-sensitive liposomes, meaning that the release was faster [56,57].

### 3.6. Stability of the Co-Delivery System PTX-TSL (siCOX-2(9R/DG-GNS))

To assess the stability of PTX-TSL-siCOX-2(9R/DG-GNS), samples were measured over different time periods at 4 °C. The particle sizes were 294.90 ± 7.21 nm, 294.75 ± 9.04 nm, and 295.63 ± 9.68 nm, and the PDI values were 0.14 ± 0.03, 0.16 ± 0.05, 0.17 ± 0.06, respectively. The minimal changes on particle size and PDI of PTX-TSL-siCOX-2(9R/DG-GNS) indicated that the co-delivery system was stable and did not aggregate at 4 °C for three weeks.

### 3.7. In Vitro Formulation Compatibility

MTT assay was used to evaluate the cytotoxicity of free GNS and GNS complexes on HUVECs. As shown in Figure 4C, the viability of the HUVECs after treatment with GNS, 9R-GNS, 9R/DG-GNS and TSL-(9R/DG-GNS) was approximately 72%, 76%, 80% and 82% at the highest concentration (1000 μmol/L), respectively. Notably, the viability of cells treated by TSL-(9R/DG-GNS) at any other doses (0.01, 0.1, 1, 10, 100 μmol/L) were all ≥90%. Notably, the GNS complex displayed improved biocompatibility after modification with TSL. TSL could shield the positive charge of 9R and the toxic components of GNS, which enhanced biocompatibility.

### 3.8. In Vitro Cellular Uptake

We evaluated the cellular uptake of siCOX-2 and GNS complexes in paclitaxel-resistant HepG2 cells (HepG2/PTX cells) using confocal laser-scanning microscopy (CLSM) (Figure 5). Fluorescence imaging (Figure 5A) indicated that siCOX-2(GNS) was unable to enter HepG2/PTX cells. siCOX-2(9R-GNS) could partially penetrate after 4 h incubation, while the levels of siCOX-2(9R/DG-GNS) entry were high, particularly under heating conditions (43 °C, 15 min). Only small amounts of PTX-TSL-siCOX-2(9R/DG-GNS) entered the cells in the absence of laser irradiation. However, PTX-TSL-siCOX-2(9R/DG-GNS) under heating conditions displayed high levels of cell entry, evidenced by the increased CLSM fluorescence intensity. This was further confirmed by the quantitative analysis of the averaged fluorescent intensities of each sample at a 4 h incubation time (Figure 5B). A significantly lower intracellular fluorescence intensity was found in the cells treated with the siCOX-2(GNS), compared to those of other nanoparticles. siCOX-2(9R-GNS) partially entered cells, which was related to the transmembrane effects of 9R. GNS modified with 9R and DG could be taken up by tumor cells through the targeting effects of DG and the transmembrane effects of 9R. At 43 °C, membrane permeability was higher and the levels of siCOX-2(9R/DG-GNS) entering the cells increased. The levels of endocytosed PTX-TSL-siCOX-2(9R/DG-GNS) were low. This was because the non-laser irradiation group of siCOX-2(9R/DG-GNS) was coated with TSL, which had shielding action on DG and 9R. However, when the temperature reached the phase transition temperature of TSL, PTX and internal siCOX-2(9R/DG-GNS) could be released. Hence, PTX-TSL-siCOX-2(9R/DG-GNS) entered cells to significantly higher levels under heating conditions (43 °C, 15 min).

### 3.9. Gene Silencing Efficiency of PTX-TSL-siCOX-2(9R/DG-GNS)

The gene silencing efficiency of the siCOX-2(GNS) complex in HepG2/PTX cells was assessed by a Western blot analysis. As shown in Figure 6, compared to siCOX-2(GNS) and siCOX-2/9R-GNS groups, COX-2 silencing was higher in the siCOX-2(9R/DG-GNS) group (*p* < 0.001 and *p* < 0.01 respectively). Under heating conditions (43 °C, 15 min), siCOX-2(9R/DG-GNS) could significantly downregulate the expression of COX-2 expression (*p* < 0.05). This was likely due to the DG targeting effects [58,59,60], the transmembrane effects of 9R, and increased heat-induced membrane permeability [61,62]. Increased temperatures can lead to increases in cell membrane permeability and enhance the transfection efficiency of gene delivery systems. siCOX-2(9R/DG-GNS) complexes were encapsulated by PTX-TSL but did not downregulate COX-2. COX-2 silencing in HepG2/PTX cells increased after treatment with PTX-TSL-siCOX-2(9R/DG-GNS), most notably at 43 °C. In the presence of heating, TSL membranes were destroyed and siCOX-2(9R/DG-GNS) was exposed, promoting PTX and siCOX-2 release. This explains why after heating, PTX-TSL-siCOX-2(9R/DG-GNS) could significantly silence COX-2 expression.

### 3.10. Effects of PTX-TSL-siCOX-2(9R/DG-GNS) on Cancer Cell Growth

To assess the effects of PTX-TSL-siCOX-2(9R/DG-GNS), drug-resistant cell growth experiments were performed. Figure 7A shows that increasing siCOX-2 concentrations led to enhanced antitumor effects. Compared to siCOX-2(GNS) and siCOX-2(9R-GNS), siCOX-2(9R/DG-GNS) more potently inhibited HepG2/PTX cell growth (*p* < 0.05). When siCOX-2(9R/DG-GNS) was encapsulated with PTX-TSL for combination therapy, its antitumor effects significantly improved (*p* < 0.05). Moreover, the antitumor effects of each group increased at 43 °C (15 min). At the maximum concentration of siCOX-2 (100 nM), the viability of PTX-TSL-siCOX-2(9R/DG-GNS) treated HepG2/PTX cells was as low as 23% (*p* < 0.001). These enhanced effects were most likely due to the ability of siCOX-2 to reverse tumor resistance following heating, which enhanced both transfection efficiencies, gene silencing, and promoting tumor cell apoptosis [63,64,65].

Co-delivery systems combining gene therapeutic agents (miRNA and siRNA) and anti-tumor drug are highlighted as effective and innovative systems for overcoming multidrug resistance (MDR) [66,67]. Physical approaches, such as the combination of drug with photodynamic/ultrasound /thermal therapies to overcome MDR have also been focused on [68,69,70,71]. In contrast, the nanoplatform based gene-, chemo- and photothermal combination therapy for reversing MDR was rarely studied [72]. Our research shows that chemo- and gene- co-delivery system delivered with hybrid gold nanostars coated by temperature sensitive liposomes (PTX-TSL-siCOX-2(9R/DG-GNS)) could improve paclitaxel-resistance in hepatic carcinoma in combination with thermal therapy.

### 3.11. Cell Apoptosis

Annexin V-FITC/PI double staining was performed to determine the apoptosis of HepG2/PTX cells treated by different nanoparticles. As shown in Figure 7B, compared to siNC(9R/DG-GNS) group, the cells treated with siCOX-2(9R/DG-GNS) under 37 °C displayed a significantly higher rate of apoptosis, ~20.8% (sum of late apoptosis percentage (Q2) and early apoptosis percentage (Q3)). These results can be attributed to the combination of 9R/DG-GNS and siCOX-2. Compared to siCOX-2(9R/DG-GNS) group, the cells treated with PTX-TSL-siCOX-2(9R/DG-GNS) also displayed notable apoptosis, with apoptosis rates of ~34.8%. These results can be mostly attributed to the chemotherapeutic effects of PTX and the existing of TSL, which can protect the stability of the encapsulated siCOX-2. In addition, the cells treated with PTX-TSL-siCOX-2(9R/DG-GNS) followed by 43 °C heating for 15 min had apoptosis rate increased to ~47%. Upon heating, both the increased temperature and the enhanced drug release contributed to the induction of cell apoptosis. These results confirm that the hyperthermia combined with chemotherapy and gene therapy could significantly increase the cell growth inhibition rate.

## 4. Conclusions

In summary, we successfully prepared siCOX-2 loaded GNS composite nanoparticles siCOX-2(9R/DG-GNS). The PTX-TSL-siCOX-2(9R/DG-GNS) co-delivery system was further constructed. The delivery system has many advantages, including rapid release and rapid uptake by tumor cells under elevated temperatures. The results of the anti-tumor drug resistant assays showed that siCOX-2 could effectively overcome resistance to PTX. Additionally, drug-resistant cancer cells could be targeted by the photothermal conversion effects of GNS. Therefore, the PTX-TSL-siCOX-2(9R/DG-GNS) system holds promise for the treatment of anti-tumor drug resistance.

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
