# Peer review of "Combined Modality Therapy Based on Hybrid Gold Nanostars Coated with Temperature Sensitive Liposomes to Overcome Paclitaxel-Resistance in Hepatic Carcinoma"

_pharmaceutics, 2019, doi:10.3390/pharmaceutics11120683_

Round 1

Reviewer 1 Report

In the manuscript entitled "Combined modality therapy based on hybrid gold nanostars coated with temperature sensitive liposomes to overcome paclitaxel-resistance in hepatic carcinoma" the authors bring a significant amount of innovative and relevant data. However, some mayor points should be evaluated before consideration for publication. 

I would highlight as a mayor flaw, the lack of discussion. The authors should bring more literature that corroborates their work and connect it to similar research that has been done, showing why their system brings something new into light. 

Additionally:

On line 188, when reported that " the percentage of the drug in the samples calculated", authors are encouraged to add the formula, so that it is more clear to the reader.  On items 2.6.1 and 2.6.4: Authors should report cell densities platted in 96-well plates On item 2.6.2 Authors say cells were transfected with the formulations. Specify that they were transfected with Carbofluorescein labeled RNA and  treated with formulations.  On line 380, authors write "At decreasing drug concentrations", change for at decreasing formulation concentrations, as no drugs are present. Authors are also suggested to change the item 3.7 title to "in vitro formulation compatibility" so that it does not confuse the reader On line 301 authors state : " Through DLS, the sizes of the GNS composite carriers were larger than those observed by TEM, most likely as the result of water evaporation during TEM sample preparation". Authors are encouraged to better explain that this is due to the hydrodinamic radius that is measured by DLS, and add a citation that corroborates their finding making the study more sound. On figure 2 items C, D and E it is pretty clear that scale bars do not match the sizes described in the text. Please do check where the mistake is. On item 3.5 authors report " The peak area of PTX-TSL was measured as209734.46" Please clarify if this is an average result of how many samples or how do you think this value is trustfull. Regarding the stability study: Study design is extremely poor. Authors evaluated only the particle sizes and PDI during different time points. These evaluations do not allow any inferences about what happens to drug loading during time. PTX is usually pretty unstable in lipidic bilayers, therefore, authors should present the HPLC quantification of the PTX for the different evaluated time points.  Authors must include a topic "2.5.6 Statistical analysis" describing which kind of statistical analyses were used.

Concerning some formatting and language aesthetic:

In the materials, list origin as (city, state, country). Please also standardize it on item 2.5.2.

Avoid redundancy in line 250: "preparation process were
schematically illustrated in Scheme 1". Write only "preparation process were
illustrated in Scheme 1".

Also in line 260 : "Scheme 1. Schematic overview of the preparation...". Write as "Scheme 1. Overview of the preparation.."

Along the text authors refer as PTX-resistance Cells. The right term is PTX-resistant ! Please correct along text.

Line 234: where written "the cells were collection" write "the cells were collected"

Reviewer 2 Report

This is an interesting manuscript where the authors have done a lot of work to prove the superiority of their system over the existing ones. The complexity of materials used to make a delivery system that is better that the others is appreciable. While this manuscript has a lot of data it lacks the attention to detail that needs to be given in order to make it perfect for publication yet. I recommend that the authors consider some of the comments mentioned below prior to resubmitting the article for consideration.
Results and Discussion:
In Scheme 1, none of the acronyms are defined. Without the explanation of what each of those are, it is extremely hard to evaluate the process taking place. What are DPPC, MSPC, DSPE. While these have been mentioned in the main text, It is important to be mentioned in the figure legend also.
Line 250 to 258 - The authors list out ppm values of all the groups present in the DG-PEG-Cys and H3CO-PEG-Cys composite. While the ppm values for individual groups for the listed materials is correct, I do not see any information regarding how the presence of these groups directly correspond to the bond between the available components. The individual materials when put into an NMR would give out the exact same values as the ones shown in the supplementary data. Every individual material used in this experiments needs to analyzed using NMR first. The composite material should then be analyzed to show a change in the bonding groups when an NMR is performed. The material in its current form is unacceptable to be considered a covalently bound composite.
Lines 263-276: None of the FTIR results show that there seems to be a change in the bond formation either. Both NMR and FTIR results point towards no covalent bond formation between the groups of DG-PEG-Cys. The C-O-C stretching for 9R/DG-PEG compared
Figure2: I did not understand why did the particle size of gold increase when other materials were added onto coat the GNS. I would imagine that the GNS would remain to be of same size but transparent coatings around it would increase in size. Correct me if I am wrong, but all the images from A to E look like they are all GNS. Any spot EDS to confirm your findings?
Figure3: (A) These are visible spectrum  findings. The authors mention that it is UV spectrum. please correct. Or make a correction that these are UV-Vis Spectrum findings. Are there any heating graphs for 9R, DG, PEG alone also. While I do not suspect them to increase the temperature of the liquid they are present in , it would serve as a good negative control.
Section 3.4: What does binding ability mean. The hypothesis of conducting this experiment was not properly explained in the manuscript. What does it mean, what result are you expecting, what result would mean to be positive and negative needs to be explained. Putting up a reference to a previously conducted study would also help while mentioning the importance of the assay.
Figure 6: The full blot for the western blot conducted in this figure needs to be shown in the supplementary data. The blot seems to be slightly angled and the full blot will provide better understanding. Also mention the Mol Wt of the proteins being tested. Also mention the companies from which the antibodies were purchased given that the concentrations of the Abs tend to vary between suppliers. Which secondary antibodies were used and at what concentration.
Line 232: What does 25 50 100nM 7nM mean? Are all the concentrations in nM or only the final ones in nM? The protocol mentioned for MTT assay is incomplete and causes confusion for people trying to replicate. Refer to a previous article that performed MTT assay and mention that the procedure was similar to that.
In the manuscript, what kind of statistical analysis was performed in Fig 5, 6 and 7? Please mention a statistical analysis section in the methods section. Stats need to be performed for Fig 4B, 4C.
Figure 7: (A) There seems to be significant decrease in cell viability in cells that did not receive the heat treatment. Isn't this against what the authors claim? Although the authors claim that there is a further decrease in the cell viability in heated groups, no stats were performed between these groups to confirm this finding. (D) What are the graphs a,b,c....g? They have to be either mentioned in the figure or they need to be properly labeled in the figure legend. What are the %values for f and g that make them different from others. May be a bar chart showing the difference in these values with a statistical significance test can provide better insight. The X and Y axis values are not visible in the figures provided. They need to be larger to be able to understand the fluorescence intensities.
Line 243, check the sentence formation, "the cells were collection"

Round 2

Reviewer 1 Report

Manuscript was significantly improved. 

Authors should justify the inclusion of a new author in this stage of the process. New author initials on affiliation should be checked. 

Reviewer 2 Report

Accept